# Novel approaches in linkage of data sources to explore the associations between purchase of opioid prescriptions during pregnancy and adverse neonatal outcomes

Nahed O. ElHassan[1], Corey J. Hayes[2], Ruchira V. Mahashabde[2], Xiaotong Han[3], Chary Akmyradov[4], Chenghui Li[2], Teresa Hudson[5], Robert Mcgehee Jr[1,6], Peter M. Mourani[4,7¤], Bradley C. Martin[2]*

1 Department of Pediatrics, Division of Neonatal-Perinatal Medicine, Arkansas Children's Hospital (ACH), University of Arkansas for Medical Sciences (UAMS), Little Rock, Arkansas, United States of America, 2 College of Pharmacy, Division of Pharmaceutical Evaluation and Policy, UAMS, Little Rock, Arkansas, United States of America, 3 Department of Psychiatry, Center for Health Services Research, UAMS, Little Rock, Arkansas, United States of America, 4 Arkansas Children's Research Institute, Little Rock, Arkansas, United States of America, 5 Department of Emergency Medicine, UAMS, Little Rock, Arkansas, United States of America, 6 Graduate School, UAMS, Little Rock, Arkansas, United States of America, 7 Department of Pediatrics, Division of Critical Care Medicine, ACH, UAMS, Little Rock, Arkansas, United States of America

¤ Current Address: Department of Pediatrics, Division of Critical Care Medicine, Ann Arbor, Michigan, USA
* bmartin@uams.edu

## Abstract

### Objective

To evaluate patterns of opioid purchases by payment source among pregnant women in Arkansas and examine their associations with adverse neonatal outcomes.

### Methods

Liveborn singleton infants born in Arkansas between 2014 and 2016 were identified from the Birth Certificate Records and linked to the All-Payer Claims Database (APCD), capturing public and private insurance claims; the Prescription Drug Monitoring Program (PDMP), recording controlled substance dispensations regardless of payment source; and the Social Determinants of Health Database, providing neighborhood socioeconomic indicators. Pregnancies were categorized as opioid non-buyers or buyers (insurance-only or self-paid). Adjusted odds ratios (AORs) for neonatal outcomes were estimated using Generalized Estimating Equations. Trimester-specific opioid exposure, expressed per 100 morphine milligram equivalents (MME), was analyzed to assess time-related effects.

### Results

This longitudinal retrospective cohort analysis included 27,441 pregnancies; 21% (5,841) involved opioid purchases, with 15% (880) including at least one self-paid

**Data availability statement:** Data used in this study were obtained from the Arkansas All-Payer Claims Database (APCD), which is governed by state and federal privacy protections and cannot be shared publicly under the terms of the Data Use Agreement with the Arkansas Center for Health Improvement (ACHI) and the Arkansas Insurance Department. Interested researchers may request access to APCD data for approved research projects through the formal request process at www.arkansasapcd. net. All analytic procedures, variable definitions, and modeling specifications are provided in the paper to support reproducibility.

**Funding:** All funding for this study was provided by the National Center for Advancing Translational Sciences of the National Institutes of Health under Award Number UL1 TR003107. The content is solely the responsibility of the authors and does not necessarily represent the official views of the National Institutes of Health. There was no additional external funding received for this study.

**Competing interests:** Dr. Martin receives royalties from TrestleTree LLC for the commercialization of an opioid risk prediction tool, which is unrelated to the current investigation and hence TrestleTree played no role in this manuscript. The remaining authors declare no financial relationships, employment, consultancy, patents, products in development, or marketed products relevant to this work.

**Abbreviations:** AMD, Adjusted mean difference; AOR, adjusted odds ratio; APCD, All Payer Claims Database; AR, Arkansas; BCR, Birth Certificate Records; CI, Confidence interval; GEE, Generalized Estimating Equations; GPI, Generic Product Identifier; ICD-CM, International Classification of Diseases, Clinical Modifications; LBW, Low birth weight; MME, Morphine milligram equivalents; NDC, National Drug Code; NICU, Neonatal intensive care unit; NOWS, Neonatal opioid withdrawal syndrome; PDMP, Prescription Drug Monitoring Program; PT, Preterm; RUCA, Rural-urban commuting area; SD, Standard deviation; SDif, Standardized difference; SDOH, Social Determinant of Health Database; SGA, Small for gestational age; VIF, Variance inflation factor.

transaction. Opioid prescription counts during pregnancy were 3,434 (PDMP), 2,161 (APCD), and 8,579 (both). Compared with non-buyers, opioid buyers had higher adjusted odds of preterm (PT) birth (AOR 1.14, 95% CI 1.02–1.27) and neonatal opioid withdrawal syndrome (NOWS) (AOR 2.10, 95% CI 1.42–3.11), with no significant associations observed for low birth weight, NICU admission, or birth-weight z-score. Increasing MME was associated with higher odds of NOWS in first (AOR 1.0045, 95% CI 1.0011–1.0080), second (AOR 1.0064, 95% CI 1.0025–1.0103), and third (AOR 1.0206, 95% CI 1.0122–1.0291) trimester. No significant differences were found between insurance-only and self-paid buyers for any neonatal outcome.

## Conclusions

Linking APCD and PDMP enabled a comprehensive assessment of prescribed opioid exposure. Opioid purchases were associated with increased risk of NOWS and modestly higher odds of PT birth. The dose and timing of opioid purchase were key determinants of NOWS. Payment source was not associated with differences in neonatal risk.

## Introduction

Over the past three decades, the use of psychoactive drugs has increased among pregnant women in the U.S [1, 2]. Maternal opioid use, in particular, is a growing public health concern due to its potential adverse effects on both the mother and developing fetus. Opioid misuse during pregnancy has been associated with neonatal opioid withdrawal syndrome (NOWS) and higher risks of preterm (PT) delivery, small for gestational age (SGA), low birth weight (LBW), and neonatal intensive care unit (NICU) admission [3]. However, many studies assessing neonatal outcomes following maternal opioid use have been limited by exposure misclassification bias, often relying on self-reported surveys that likely underestimate actual opioid exposure [4]. Others failed to evaluate differences in maternal characteristics between opioid users and non-users [3].

To address the opioid crisis, the Centers for Disease Control and Prevention identified Prescription Drug Monitoring Programs (PDMPs), state-run electronic databases that track controlled substance prescriptions, as a key tool for reducing opioid misuse [5]. Implemented in 2013, the Arkansas (AR)-PDMP tracks all opioids purchased within the state, including self-paid transactions that might evade detection in insurance claims or bypass insurer-imposed quantity restrictions [6]. However, PDMPs do not capture opioids dispensed across state lines and lack diagnostic information, limiting researchers' ability to determine medical indications for prescribed opioids.

A linkage was established between the AR All-Payer Claims Database (AR-APCD), a statewide claims database, and the AR-PDMP, 2014–2016 [7]. This study leveraged this linkage to characterize opioid prescription purchases among pregnant women and evaluate their impact on adverse neonatal outcomes. We hypothesized that pregnant women who self-paid for prescribed opioid analgesics,

potentially to bypass payer restrictions or detection, represent a higher-risk group due to higher opioid purchase, and that both opioid purchase and opioid amount are associated with increased risk of adverse neonatal outcomes (PT birth, SGA, LBW, NICU admission, and NOWS).

## Materials and methods

### Data sources

**Arkansas All Payer Claims Database (AR-APCD).** The AR-APCD is a statewide repository of claims from public and private payer types linked to eligibility and provider files [8]. This database contains demographic information, payer types, diagnosis and procedure codes, provider and dispenser information, prescription drug details, and dates of service. The AR-APCD does not capture prescriptions not submitted for reimbursement.

**Arkansas Prescription Drug Monitoring Program (AR-PDMP).** Dispensers are mandatorily required to report all controlled substances dispensed in AR, irrespective of payment source, to the AR-PDMP [9]. Data captured by AR-PDMP include medication name and strength, date dispensed, quantity, days of supply, payment source, patient name, and names of prescriber and pharmacy.

**Birth Certificate Records (BCR).** Each BCR represents a single livebirth in AR and enables the identification of maternal-infant dyads [10]. The BCR includes perinatal variables and captures data on maternal characteristics, obstetrical diagnoses, and neonatal birth outcomes.

**Agency for Healthcare Research and Quality Social Determinant of Health Database (SDOH).** This national database provides variables across five key SDOH domains: social context, economic context, education, physical infrastructure, and healthcare context, offering insights into the maternal neighborhood's social environment (S1 Appendix) [11].

### Data linkage

The AR-APCD Hash ID is a 44-character unique identifier based on an individual's last name and date of birth. When combined with the individual's sex, the Hash ID ensures unique identification for over 97% of the state population, allowing linkage within the APCD and across other databases for longitudinal analyses at the individual level without using personally identifiable information [8]. The Hash ID was used to link APCD files with BCRs and AR-PDMP. In some cases, mothers or infants in the BCR had a missing or invalid Hash ID (e.g., duplicates), preventing linkage to APCD files and were excluded. Data from the SDOH domains were linked using the 5-digit ZIP code of the maternal address at the time of delivery.

### Study design and population

This longitudinal retrospective cohort study included liveborn singleton pregnancies delivered in AR between January 1, 2014, and December 31, 2016. Non-singleton pregnancies were excluded due to the known increased risk of adverse neonatal outcomes [12,13]. To minimize bias from incomplete data, fully ascertain exposure (i.e., prescribed opioid purchases), and ensure comprehensive capture of prescription fills, maternal diagnoses, and healthcare utilization, mothers were required to have continuous medical and pharmacy benefit enrollment from six months before conception through delivery [14]. Pregnancies were further restricted to those with estimated conception dates between April 1, 2014, and March 12, 2016, to allow complete assessment of opioid purchases during the three months before conception and throughout pregnancy, while ensuring that both term and preterm deliveries were captured through the end of 2016. Eligible pregnancies were also required to have at least one medical or pharmacy claim during the coverage period to confirm healthcare utilization. Additional exclusions included gestational age < 22 or >42 weeks, implausible interpregnancy intervals (estimated conception dates that were less than 28 days after a preceding delivery), maternal age exceeding the upper limits of the reproductive range (>55 years), maternal residence outside of AR at delivery, or receipt of implausible

prescription records (opioid purchase quantities exceeding 1,000 tablets). Small variations in analytic sample sizes across outcomes reflected differences in data availability, as NOWS could only be assessed among infants with complete maternal–infant linkage to APCD medical claims, and SGA could only be calculated for infants born between 23 and 41 weeks of gestation based on the U.S. national growth curves [15].

The estimated conception date was based on gestational age at delivery recorded on BCR and date of delivery. Pregnancy trimesters were determined per the American College of Obstetricians and Gynecologists definition [16]. The study received Institutional Review Board approval at the University of Arkansas for Medical Sciences and exemption of informed consent (Number#229157). All data were de-identified, and the authors had no access to information that could identify individual participants during or after data collection. The initial linkage between APCD and PDMP data was completed in 2021 and 2022. Additional linkage to BCR was completed in 2023. Data for this project were accessed for analysis between January 1, 2024, and November 15, 2025.

## Opioid exposure and study cohorts

Opioid exposure was defined using the prescription fill date and the days supplied of the opioid analgesic (excluding buprenorphine), capturing any days overlapping with the defined trimester of pregnancy. Opioid prescriptions were identified through the National Drug Codes, enabling the calculation of morphine milligram equivalents (MME) (S1 Appendix) [17]. Duplicate prescriptions were excluded. Each prescription from the APCD and PDMP was cross-referenced based on generic drug code, days' supply, fill date, MME, and dispensing pharmacy. The study cohort was categorized into those exposed to prescribed analgesic opioids during pregnancy (opioid buyers) and those not exposed (non-buyers). Among buyers, individuals were classified into two sub-cohorts: those who exclusively used insurance to purchase opioids (insurance-only buyers) and those who self-paid for at least one opioid prescription (self-paid buyers).

## Outcome measures

The outcome measures were PT delivery (born at <37 weeks' gestation), LBW (birth weight <2500 g), SGA, and NICU admission after birth assessed using BCR. SGA was treated as a continuous outcome and defined as a birth weight z-score below 0, based on U.S. growth curve reference values by gestational age and infant sex [15] (S1 Appendix). NOWS was assessed using AR-APCD (S1 Appendix).

## Covariates

A *priori* covariates were selected per literature review to account for factors that may influence adverse neonatal outcomes among pregnant women using opioids [3,18–31]. These covariates, assessed from six months prior to pregnancy through delivery, included maternal demographic and socioeconomic characteristics, rurality of area of residence, clinical diagnoses, pharmacologic exposures, and neighborhood-level SDOH variables (S1 Appendix).

## Statistical analysis

Standardized differences (SDif) were calculated to compare characteristics between included and excluded cohorts. Descriptive statistics were presented as frequencies and proportions for categorical variables, and as means (standard deviations, SD) or medians (minimum, maximum) for continuous variables, based on data distribution. To compare baseline characteristics across groups while accounting for repeated pregnancies within mothers, Generalized Estimating Equations (GEE) were used, modeling one baseline variable at a time [32,33].

Exposure modeling proceeded in two stages. Stage 1 contrasted any opioid buyers versus non-buyers. Stage 2 evaluated a three-level exposure (non-buyers [reference], insurance-only buyers, and self-paid buyers). Multicollinearity was assessed using variance inflation factors (VIFs); all retained covariates had VIF < 5, indicating acceptable collinearity

[34]. For analyses of maternal opioid exposure and neonatal outcomes, GEE models accounted for clustering of repeated pregnancies within mothers. Models were specified with an identity link for normally distributed continuous outcomes (i.e., birth weight z-score) and a logit link for binary outcomes. Adjusted mean differences (AMDs) were reported for continuous outcomes, representing the average absolute difference in the outcome between exposure groups, whereas adjusted odds ratios (AORs) were reported for binary outcomes, representing the relative odds of the outcome after adjustment for covariates. A sequential modeling strategy was used, incrementally adjusting for demographic, geographic, clinical, and neighborhood-level variables. Robust standard errors were estimated using the sandwich variance estimator.

The association between average daily MME per 100 mg/day (MME/100) and neonatal outcomes was also assessed by trimester using an exposure × MME × trimester interaction term to examine whether timing or dose modified the effect compared with no exposure. As a post hoc exploratory analysis, potential effect modification of the exposure–outcome associations by smoking and concurrent benzodiazepine exposure was evaluated using product (interaction) terms (e.g., exposure × smoking). Because statistical power for detecting interaction effects is typically lower than for main effects, interaction terms were explored at a significance level of $P < 0.10$ to reduce the likelihood of type II error [35].

To assess the robustness of observed associations to potential unmeasured confounding, E-values were calculated for each exposure–outcome association comparing any opioid buyers versus non-buyers. E-values quantify the minimum strength of association that an unmeasured confounder would need to have with both the exposure and the outcome, conditional on measured covariates, to fully explain the observed effect [36]. All analyses were conducted using SAS version 9.4 (SAS Institute Inc., Cary, NC).

## Results

### Study cohort derivation and characteristics

Fig 1 presents the study attrition flowchart following the application of exclusion criteria, yielding a final cohort of 27,441 pregnancies among 26,771 mothers. The maternal demographic and socioeconomic characteristics of included pregnancies were largely comparable to those excluded, with SDif <|10|% [37], indicating similar distributions in maternal age, prenatal care utilization, body mass index, education level, and payer type. However, included pregnancies had a higher proportion of mothers who were non-Hispanic White (SDif = –0.2291), unmarried (SDif = 0.2021), eligible for the Special Supplemental Nutrition Program for Women, Infants, and Children (SDif=−0.1052), and residing in the southern region of AR (SDif=−0.1292) (data not shown).

Opioids were purchased in 21% (5,841 of 27,441) of included pregnancies, with 15% (880 of 5,841) involving at least one self-paid prescription. Compared to non-buyers, opioid buyers were more socioeconomically disadvantaged, had higher rates of chronic medical conditions and pain-related diagnoses, and were more likely to fill prescriptions for psychoactive medications (Table 1). They also lived in neighborhoods with greater disadvantage across the five key SDOH domains (S1 Table). Among opioid buyers, self-paid in comparaison to insurance-only opioid buyers were more often non-Hispanic White and lived in neighborhoods with a higher proportion of White-only households. They also had higher rates of mental health conditions, smoking, alcohol or substance abuse, injuries, and procedures, but were less likely to purchase antidepressants, benzodiazepines, and barbiturates during pregnancy. Self-paid buyers also consistently purchased higher mean MME doses during all trimesters compared to insurance-only buyers: first trimester (879 [SD: 3,308] vs. 287 [SD: 1,136]), second trimester (661 [SD: 2,736] vs. 216 [SD: 1,053]), and third trimester (444 [SD: 1,882] vs. 140 [SD: 701]) (all $P < 0.0001$) (Table 1).

### Comparison of opioid prescriptions in PDMP and APCD

Mothers in the study cohort filled a total of 14,174 opioid analgesic scripts. Of these, 8,579 (60.53%) were identified from both data sources, while 2,161 (15.25%) were found only in APCD and 3,434 (24.23%) only in PDMP. Many prescriptions,

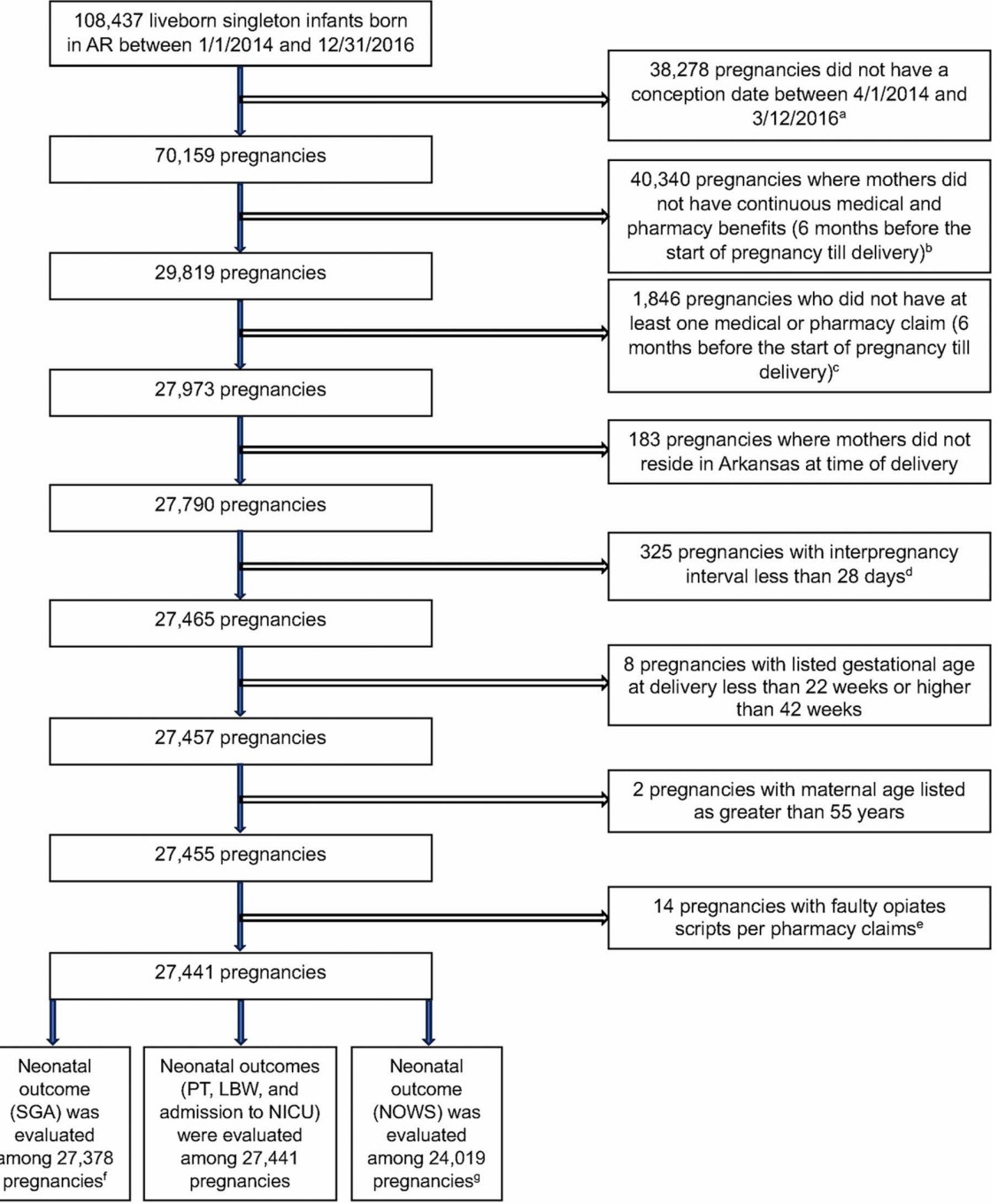

**Fig 1. Subject Attrition Diagram.** Abbreviations by alphabetical order: LBW, low birth weight; NICU, neonatal intensive care unit; NOWS, neonatal opioid withdrawal syndrome; PT, preterm; SGA, small for gestational age. [a]Pregnancies were included if conception occurred on or after 4/1/14 (allowing for a pre-pregnancy window—1/1/14 to 3/31/14—to assess opioid purchases before pregnancy) and on or before 3/12/16 (ensuring sufficient time for preterm or term delivery by 12/31/16); [b]Pregnancies were required to have continuous medical and pharmacy benefit enrollment from six months before conception through delivery to ensure complete capture of prescribed opioid exposure and relevant maternal covariates; [c]Some pregnancies were excluded since they did not have at least one medical or pharmacy claim during the coverage period to confirm healthcare utilization; [d]Pregnancies belonging to the same woman were excluded if the interval between pregnancies was implausible; [e]Scripts with abnormally high filled quantity (>1,000 tablets) were excluded; [f]SGA outcomes were not available for infants born at 22 or 42 weeks, as the U.S. national growth curve by Olsen et al. [15] provides reference data only for infants born between 23 and 41 weeks of gestation; [g]NOWS outcome could not be assessed on a subset of infants who had missing or duplicate Hash ID and could not be linked to the APCD medical claims.

**Table 1. Comparison of characteristics between the study cohorts and sub-cohorts.**

| Variable | Opioid Buyers | | | | Opioid Non-buyers N = 21600 (78.71%) | **P-value |
|---|---|---|---|---|---|---|
| | Self-Paid N = 880 (15.07%) | Insurance-Only N = 4961 (84.93%) | *P-value | Total N = 5841 (21.29%) | | |
| *Demographic* | | | | | | |
| Maternal age group, N (%) | | | 0.0274 | | | <0.0001 |
| <20 years | 56 (6.36%) | 415 (8.37%) | | 471 (8.06%) | 2479 (11.48%) | |
| ≥20–<30 years | 586 (66.59%) | 3092 (62.33%) | | 3678 (62.97%) | 12663 (58.63%) | |
| ≥30 years | 238 (27.05%) | 1454 (29.31%) | | 1692 (28.97%) | 6458 (29.99%) | |
| Race/ethnicity, N (%) | | | <0.0001 | | | <0.0001 |
| Non-Hispanic White | 615 (69.89%) | 3353 (67.59%) | | 3968 (67.93%) | 14665 (67.89%) | |
| Non-Hispanic Black | 195 (22.16%) | 1380 (27.82%) | | 1575 (26.96%) | 5154 (23.86%) | |
| Other | 70 (7.95%) | 228 (4.60%) | | 298 (5.10%) | 1781 (8.25%) | |
| Marital status (yes), N (%) | 339 (38.52%) | 2080 (41.93%) | 0.0588 | 2419 (41.41%) | 10961 (50.75%) | <0.0001 |
| Prenatal care, N (%) | | | <0.0001 | | | 0.1765 |
| Yes | 781 (88.75%) | 4345 (87.58%) | | 5126 (87.76%) | 18775 (86.92%) | |
| No | 33 (3.75%) | 86 (1.73%) | | 119 (2.04%) | 445 (2.06%) | |
| Unknown | 66 (7.50%) | 530 (10.68%) | | 596 (10.20%) | 2380 (11.02%) | |
| Eligibility for WIC (yes), N (%) | 553 (62.84%) | 3150 (63.50%) | 0.7104 | 3703 (63.40%) | 11390 (52.73%) | <0.0001 |
| Pre-pregnancy body mass index, N (%) | | | 0.0002 | | | <0.0001 |
| <=18 | 90 (10.23%) | 323 (6.51%) | | 413 (7.07%) | 1556 (7.20%) | |
| 19-24 | 289 (32.84%) | 1823 (36.75%) | | 2112 (36.16%) | 8593 (39.78%) | |
| 25-29 | 211 (23.98%) | 1097 (22.11%) | | 1308 (22.39%) | 4935 (22.85%) | |
| >=30 | 290 (32.95%) | 1718 (34.63%) | | 2008 (34.38%) | 6516 (30.17%) | |
| Geographic location of maternal residence, N (%) | | | 0.0011 | | | 0.1001 |
| North | 489 (55.57%) | 2463 (49.65%) | | 2952 (50.54%) | 10975 (50.81%) | |
| Central | 214 (24.32%) | 1492 (30.07%) | | 1706 (29.21%) | 6504 (30.11%) | |
| South | 177 (20.11%) | 1006 (20.28%) | | 1183 (20.25%) | 4121 (19.08%) | |
| Level of education, N (%) | | | 0.0051 | | | <0.0001 |
| Less than high school | 173 (19.66%) | 798 (16.09%) | | 971 (16.62%) | 2887 (13.37%) | |
| High school | 338 (38.41%) | 1844 (37.17%) | | 2182 (37.36%) | 7040 (32.59%) | |
| More than high school | 360 (40.91%) | 2290 (46.16%) | | 2650 (45.37%) | 11565 (53.54%) | |
| Missing | 9 (1.02%) | 29 (0.58%) | | 38 (0.65%) | 108 (0.50%) | |
| Payer type, N (%) | | | <0.0001 | | | <0.0001 |
| Medicaid | 543 (61.70%) | 2337 (47.11%) | | 2880 (49.31%) | 8795 (40.72%) | |
| Private insurance | 254 (28.86%) | 2247 (45.29%) | | 2501 (42.82%) | 11042 (51.12%) | |
| Other | 83 (9.43%) | 377 (7.60%) | | 460 (7.88%) | 1763 (8.16%) | |
| *Rurality of maternal area of residence* | | | | | | |
| Primary Rural-Urban Commuting Area Code, N (%) | | | 0.407 | | | <0.0001 |
| Metropolitan | 518 (58.86%) | 3007 (60.61%) | | 3525 (60.35%) | 13552 (62.74%) | |
| Micropolitan | 188 (21.36%) | 1038 (20.92%) | | 1226 (20.99%) | 4336 (20.07%) | |
| Small town | 128 (14.55%) | 715 (14.41%) | | 843 (14.43%) | 2673 (12.38%) | |
| Rural | 46 (5.23%) | 201 (4.05%) | | 247 (4.23%) | 1039 (4.81%) | |
| *Maternal medical diagnosis* | | | | | | |
| Mental health disorders (yes), N % | | | <0.0001 | | | <0.0001 |
| 0 | 500 (56.82%) | 3092 (62.33%) | | 3592 (61.50%) | 17367 (80.40%) | |
| 1 | 131 (14.89%) | 806 (16.25%) | | 937 (16.04%) | 2086 (9.66%) | |
| >=2 | 249 (28.30%) | 1063 (21.43%) | | 1312 (22.46%) | 2147 (9.94%) | |

*(Continued)*

| Variable | Opioid Buyers | | | | Opioid Non-buyers N = 21600 (78.71%) | **P-value |
|---|---|---|---|---|---|---|
| | Self-Paid N = 880 (15.07%) | Insurance-Only N = 4961 (84.93%) | *P-value | Total N = 5841 (21.29%) | | |
| Substance or alcohol abuse, (yes), N % | 281 (31.93%) | 1076 (21.69%) | <0.0001 | 1357 (23.23%) | 2530 (11.71%) | <0.0001 |
| Smoking (yes), N (%) | 468 (53.18%) | 2056 (41.44%) | <0.0001 | 2524 (43.21%) | 5431 (25.14%) | <0.0001 |
| Diabetes (yes), N (%) | 117 (13.30%) | 703 (14.17%) | 0.4910 | 820 (14.04%) | 2342 (10.84%) | 0.0001 |
| Placental pathologies (yes), N (%) | 67 (7.61%) | 367 (7.40%) | 0.8219 | 434 (7.43%) | 1422 (6.58%) | 0.0240 |
| Previous cesarean section (yes), N (%) | 218 (24.77%) | 1179 (23.77%) | 0.5185 | 1397 (23.92%) | 4041 (18.71%) | <0.0001 |
| Chronic hypertension or pregnancy induced hypertension (yes), N (%) | 164 (18.64%) | 871 (17.56%) | 0.4396 | 1035 (17.72%) | 2469 (11.43%) | <0.0001 |
| Preeclampsia with or without eclampsia (yes), N (%) | 103 (11.70%) | 678 (13.67%) | 0.1150 | 781 (13.37%) | 2556 (11.83%) | 0.0016 |
| Acute or chronic renal disease or pregnancy related renal disease (yes), N (%) | 29 (3.30%) | 122 (2.46%) | 0.1497 | 151 (2.59%) | 187 (0.87%) | <0.0001 |
| Cardiac disease (yes), N (%) | 46 (5.23%) | 186 (3.75%) | 0.0385 | 232 (3.97%) | 504 (2.33%) | <0.0001 |
| Chorioamnionitis or other maternal infection excluding HIV (yes), N (%) | 118 (13.41%) | 559 (11.27%) | 0.0674 | 677 (11.59%) | 2059 (9.53%) | <0.0001 |
| Sepsis or shock (yes), N (%) | 12 (1.36%) | 29 (0.58%) | 0.0107 | 41 (0.70%) | 60 (0.28%) | 0.0002 |
| Gastrointestinal disease (yes), N (%) | 345 (39.20%) | 1764 (35.56%) | 0.0379 | 2109 (36.11%) | 3655 (16.92%) | <0.0001 |
| Pulmonary pathologies (yes), N (%) | 111 (12.61%) | 534 (10.76%) | 0.1067 | 645 (11.04%) | 1293 (5.99%) | <0.0001 |
| Hematological pathologies (yes), N (%) | 217 (24.66%) | 1039 (20.94%) | 0.0134 | 1256 (21.50%) | 3577 (16.56%) | <0.0001 |
| Other maternal diagnoses (yes), N (%) | 23 (2.61%) | 142 (2.86%) | 0.6815 | 165 (2.82%) | 295 (1.37%) | <0.0001 |
| *Maternal painful conditions* | | | | | | |
| Injuries (yes), N (%) | 349 (39.66%) | 1713 (34.53%) | 0.0033 | 2062 (35.30%) | 3226 (14.94%) | <0.0001 |
| Arthropathies/connective tissue disorders (yes), N (%) | 248 (28.18%) | 1398 (28.18%) | 0.9990 | 1646 (28.18%) | 2498 (11.56%) | <0.0001 |
| Headache/migraine, N (%) | 270 (30.68%) | 1466 (29.55%) | 0.4986 | 1736 (29.72%) | 2701 (12.50%) | <0.0001 |
| Neuro-(pathies/- muscular) disease (yes), N (%) | 80 (9.09%) | 364 (7.34%) | 0.0704 | 444 (7.60%) | 682 (3.16%%) | <0.0001 |
| Maternal procedures (yes), N (%) | 169 (19.20%) | 516 (10.40%) | <0.0001 | 685 (11.73%) | 1820 (8.43%) | <0.0001 |
| *Maternal painful symptoms* | | | | | | |
| Any painful symptoms (yes), N (%) | | | 0.1358 | | | <0.0001 |
| 0 | 234 (26.59%) | 1321 (26.63%) | | 1555 (26.62%) | 12105 (56.04%) | |
| 1 | 228 (25.91%) | 1436 (28.95%) | | 1664 (28.49%) | 6053 (28.02%) | |
| >=2 | 418 (47.50%) | 2204 (44.43%) | | 2622 (44.89%) | 3442 (15.94%) | |
| *Maternal purchase of pharmacological agents (other than opiates analgesics)* | | | | | | |
| Days of buprenorphine purchase, median [min, max] | 0 [0, 205] | 0 [0, 400] | 0.0060 | 0 [0, 400] | 0 [0, 461] | 0.0646 |
| Benzodiazepines (yes), N (%) | 22 (2.50%) | 256 (5.16%) | 0.0006 | 278 (4.76%) | 355 (1.64%) | <0.0001 |
| Selective serotonin reuptake inhibitor pharmacological agents (yes), N (%) | 32 (3.64%) | 442 (8.91%) | <0.0001 | 474 (8.12%) | 843 (3.90%) | <0.0001 |
| Tricyclic antidepressants pharmacological agents other than SSRI (yes), N (%) | 22 (2.50%) | 266 (5.36%) | 0.0003 | 288 (4.93%) | 427 (1.98%) | <0.0001 |
| Gamma-aminobutyric acid pharmacological agents (GABA) (yes), N (%) | 3 (0.34%) | 20 (0.40%) | 1.000 | 23 (0.39%) | 4 (0.02%) | <0.0001 |
| Amphetamine pharmacological agents (yes), N (%) | 7 (0.80) | 64 (1.29%) | 0.2172 | 71 (1.22%) | 152 (0.70%) | 0.0009 |
| Antipsychotic pharmacological agents (yes), N (%) | 10 (1.14%) | 71 (1.43%) | 0.4907 | 81 (1.39%) | 104 (0.48%) | <0.0001 |
| Barbiturate pharmacological agents (yes), N (%) | 24 (2.73%) | 285 (5.74%) | 0.0002 | 309 (5.29%) | 361 (1.67%) | <0.0001 |
| Beta-Blockers pharmacological agents (yes), N (%) | 24 (2.73%) | 265 (5.34%) | 0.0010 | 289 (4.95%) | 499 (2.31%) | <0.0001 |
| Thyroid pharmacological agents (yes), N (%) | 5 (0.57%) | 57 (1.15%) | 0.1213 | 62 (1.06%) | 255 (1.18%) | 0.4783 |
| Selected antibiotics (yes), N (%) | 93 (10.57%) | 1293 (26.06%) | <0.0001 | 1386 (23.73%) | 3583 (16.59%) | <0.0001 |

*(Continued)*

**Table 1.** (Continued)

| Variable | Opioid Buyers | | | | Opioid Non-buyers N = 21600 (78.71%) | **P-value |
|---|---|---|---|---|---|---|
| | Self-Paid N = 880 (15.07%) | Insurance-Only N = 4961 (84.93%) | *P-value | Total N = 5841 (21.29%) | | |
| Morphine milligram equivalent, mean (SD) | | | | | | |
| First trimester | 879 (3308) | 287 (1136) | <0.0001 | 376 (1671) | NA | NA |
| Second trimester | 661 (2736) | 216 (1053) | <0.0001 | 283 (1447) | NA | NA |
| Third trimester | 444 (1882) | 140 (701) | <0.0001 | 186 (981) | NA | NA |

*P-value for contrast of opioid analgesic self-paid buyers versus insurance-only buyers;**P-value for contrast of all opioid analgesic buyers versus non-buyers.

NA, not applicable

regardless of when they were filled, had days of supply that overlapped pregnancy trimesters. Specifically, 6,351 prescriptions overlapped the first trimester, 4,962 overlapped the second trimester, and 3,693 overlapped the third trimester. Overlap indicates that a prescription's days of supply intersected a given trimester, whether the script was filled before pregnancy, during that trimester, or in an earlier trimester with remaining days supplied carried forward.

## Comparison of adverse neonatal outcomes

In unadjusted analyses, infants born to pregnancies in which the mother was an opioid buyer had higher rates of PT birth (10.97% vs. 8.63%, $P < 0.0001$), LBW (9.02% vs. 7.17%, $P < 0.0001$) and NICU admission (8.17% vs. 6.58%, $P < 0.0001$) compared with those whose mothers were non-buyers (Table 2). The mean birthweight z-score (SD) for opioid buyers was significantly lower than that of non-buyers relative to the reference population (−0.0690 [0.9124] vs. −0.0064 [0.9118], $P < 0.0001$). Among pregnancies linked to neonatal APCD medical claims, a higher proportion of infants born to opioid buyers were diagnosed with NOWS compared to those whose mothers did not buy opioids during pregnancy (1.67% vs. 0.38%, $P < 0.0001$). Infants of mothers who self-paid for opioids at least once during pregnancy had higher rates of NOWS than those whose mothers used insurance exclusively (3.13% vs. 1.42%, $P < 0.0001$).

## Adjusted adverse neonatal outcomes

Table 3 presents results from unadjusted and fully adjusted models, while AORs from sequential regression models comparing opioid buyers with non-buyers, insurance-only buyers with non-buyers, self-paid buyers with non-buyers, and self-paid buyers with insurance-only buyers are summarized in S2 Table. Before model fitting, multicollinearity among covariates was evaluated using VIF. Moderate correlations were observed among several neighborhood-level socioeconomic indicators and maternal education. To reduce redundancy and improve model stability, highly collinear community-level variables (e.g., percentage of Medicaid enrollment, food-stamp utilization, and limited-English households) were excluded. Maternal education was collapsed into a binary category (≤high school vs>high school), and observations with missing education data were excluded from adjusted models. After these refinements, all VIF values were <5, indicating acceptable levels of collinearity.

Opioid purchase was significantly associated with all evaluated neonatal outcomes in unadjsuted analyses. In the fully adjusted models, these associations attenuated toward the null for NICU admission (AOR 1.07, 95% CI 0.94–1.22), LBW (AOR 1.13, 95% CI 0.99–1.27), and birth-weight z-score (AMD −0.0239; 95% CI −0.0514 to 0.0037). A modest but statistically significant association persisted for PT birth (AOR 1.14, 95% CI 1.02–1.27). In contrast, opioid exposure remained strongly associated with NOWS, with fully AORs of 2.10 (95% CI 1.42–3.11) for any opioid buyer, 1.97 (95% CI 1.29–3.01) for insurance-only buyers, and 2.63 (95% CI 1.54–4.50) for self-paid buyers compared with non-buyers. No significant

**Table 2. Comparison of neonatal characteristics and outcomes between the study cohorts and sub-cohorts.**

| Variable | Opioid Buyers | | | | Opioid Non-Buyers N=21600 (78.71%) | P-value |
|---|---|---|---|---|---|---|
| | Self-Paid N=880 (15.07%) | Insurance-Only N=4961 (84.93%) | P-value | Total N=5841 (21.29%) | | |
| Male infant (yes), N (%) | 424 (48.18%) | 2583 (52.07%) | 0.0340 | 3007 (51.48%) | 11121 (51.49%) | 0.9651 |
| Mode of delivery, N (%) | | | 0.0810 | | | <0.0001 |
| Cesarean | 329 (37.39%) | 1738 (35.03%) | | 2067 (35.39%) | 6539 (30.27%) | |
| Vaginal | 529 (60.11%) | 3073 (61.94%) | | 3602 (61.67%) | 14267 (66.05%) | |
| Other | 22 (2.50%) | 150 (3.02%) | | 172 (2.94%) | 794 (3.68%) | |
| Preterm (yes), N (%) | 111 (12.61%) | 530 (10.68%) | 0.0910 | 641 (10.97%) | 1864 (8.63%) | <0.0001 |
| Low birth weight (LBW, Birth weight <2500 gm) (yes), N (%) | 88 (10.0%) | 439 (8.85%) | 0.2720 | 527 (9.02%) | 1549 (7.17%) | <0.0001 |
| Birth weight, mean z-score (SD) | −0.0802 (0.9035) | −0.0670 (0.9140) | 0.6907 | −0.0690 (0.9124) | −0.0064 (0.9118) | <0.0001 |
| Neonatal intensive care unit admission (yes), N (%) | 83 (9.43%) | 394 (7.94%) | 0.1370 | 477 (8.17%) | 1422 (6.58%) | <0.0001 |
| Neonatal opioid withdrawal syndrome (NOWS) (yes), N (%) | N=768 24 (3.13%) [a]Missing: 112 | N=4371 62 (1.42%) [a]Missing: 590 | 0.0007 | N=5139 86 (1.67%) [a]Missing: 702 | N=18880 72 (0.38%) [a]Missing: 2720 | <0.0001 |

[a]Missing values in instances where an infant's BCR could not be linked to APCP medical claim

**Table 3. Unadjusted and fully adjusted regression models evaluating associations between maternal opioid purchasing and neonatal outcomes.**

| Model | Comparison among groups | NICU Admission (yes) | LBW (yes) | PT Birth (yes) | Birth weight z-score | NOWS (yes) |
|---|---|---|---|---|---|---|
| | | AOR (95% CI) | AOR (95% CI) | AOR (95% CI) | Adjusted mean difference (95% CI) | AOR (95% CI) |
| Unadjusted Model | Opioid buyers vs. non-buyers | 1.26 (1.13-1.41)*** | 1.28 (1.15-1.42)*** | 1.31 (1.19-1.44)*** | −0.0612 (−0.0877; −0.0348)**** | 4.45 (3.22-6.15)**** |
| | Opioid insurance-only buyers vs. non-buyers | 1.23 (1.09-1.38)** | 1.25 (1.12-1.40)** | 1.27 (1.15-1.40)** | −0.0585 (−0.0869; −0.0302)**** | 3.81 (2.69-5.42)**** |
| | Opioid self-paid buyers vs. non-buyers | 1.47 (1.16-1.86)** | 1.44 (1.14-1.80)** | 1.52 (1.24-1.87)** | −0.0766 (−0.1377; −0.0154)* | 8.10 (4.96-13.22)**** |
| | Opioid self-paid vs. insurance-only buyers | 0.83 (0.65-1.07) | 0.87 (0.68-1.11) | 0.85 (0.67-1.04) | −0.0181 (−0.0832; 0.0471) | 2.12 (1.28-3.52)*** |
| Fully Adjusted Model | Opioid buyers vs. non-buyers | 1.07 (0.94-1.22) | 1.13 (0.99-1.27) | 1.14 (1.02-1.27)* | −0.0239 (−0.0514; 0.0037) | 2.10 (1.42-3.11)*** |
| | Opioid insurance-only buyers vs. non-buyers | 1.06 (0.93-1.22) | 1.12 (0.98-1.27) | 1.11 (0.99-1.25) | −0.0261 (−0.0551; 0.0029) | 1.97 (1.29-3.01)** |
| | Opioid self-paid buyers vs. non-buyers | 1.10 (0.85-1.43) | 1.17 (0.92-1.50) | 1.26 (1.01-1.58)* | −0.0107 (−0.0708; 0.0494) | 2.63 (1.54-4.50)*** |
| | Opioid self-paid vs. insurance-only buyers | 0.97 (0.74-1.26) | 0.95 (0.74-1.23) | 0.88 (0.70-1.12) | 0.0154 (−0.0470; 0.0779) | 1.34 (0.79-2.26) |

Model specifications: Unadjusted model: No maternal variables included. Fully adjusted model: Adjusted for maternal demographic characteristics, rurality of maternal residence, clinical variables (including diagnoses of painful conditions and painful symptoms), maternal purchase of pharmacologic agents other than opioid analgesics, and neighborhood-level social determinants of health.

Abbreviations (alphabetical order): AOR, adjusted odds ratio; CI, confidence interval; LBW, low birth weight; NICU, neonatal intensive care unit; NOWS, neonatal opioid withdrawal syndrome; PT, preterm birth.

Significance levels: *$P<0.05$; **$P<0.01$; ***$P<0.001$; ****$P<0.0001$

differences were observed between insurance-only and self-paid buyers for any neonatal outcome in the fully adjusted models.

## Associations between maternal characteristics and neonatal outcomes

AORs for covariates from the full models of opiate buyers with non-buyers are provided in S3 Table. Compared with non-Hispanic White women, those identified as non-Hispanic Black had higher odds of PT birth (AOR 1.29; 95% CI 1.14–1.48), NICU admission (AOR 1.17; 95% CI 1.01–1.37), and lower birth weight (AMD= −0.3345; 95% CI −0.3675 to −0.3014), but lower odds of NOWS (AOR 0.55; 95% CI 0.31–0.98).

## Sensitivity analysis for unmeasured confounding (E-Value analysis)

The robustness of adjusted associations was further assessed using E-values (Table 4). The lower confidence bounds of the E-values included 1.0 for NICU admission, LBW, and birth weight z-score, which is consistent with the null associations between prescribed opioid purchase and these outcomes. The E-value for PT birth (1.53; 95% CI 1.15–1.86) suggested modest robustness to potential unmeasured confounding. In contrast, the higher E-value for NOWS (3.63; 95% CI 2.20–5.68) indicated that only a confounder strongly associated, approximately twofold or greater, with both exposure and outcome could fully account for the observed relationship, supporting the robustness of this finding.

## Trimester-specific associations between opioid dose and neonatal outcomes

A significant interaction was detected between opioid dose and trimester of exposure, indicating that the timing of opioid use during pregnancy modified the relationship with neonatal outcomes (Table 5). Each 100 mg/day increase in average daily MME during the second trimester was associated with higher odds of NICU admission (AOR 1.0054, 95% CI 1.0010−1.0098) and LBW (AOR 1.0064, 95% CI 1.0025−1.0103), despite the overall exposure–outcome effects being nonsignificant in the fully adjusted models (Table 3). For NOWS specifically, trimester-specific dose effects were statistically significant in all three trimesters, with the magnitude of association increasing as pregnancy progressed. The strongest trimester-specific dose–response relationship was observed in the third trimester, where each 100 mg/day increase in MME was associated with a 2% higher risk of NOWS (AOR 1.0206, 95% CI 1.0122−1.0291).

## Effect modification by maternal benzodiazepine co-exposure and smoking

Interaction analyses suggested a potential modifying effect of benzodiazepine co-exposure on the association between maternal opioid use and fetal growth. The opioid × benzodiazepine term for birth-weight z-score approached statistical significance ($\chi^2 = 3.78$, $P = 0.052$) (Table 6). Stratified contrasts showed that opioid exposure was associated with a significantly lower mean birth-weight z-score among benzodiazepine buyers (AMD= −0.17 SD; 95% CI −0.32 to −0.02), whereas no difference was observed among non-buyers (−0.02 SD; 95% CI −0.05 to 0.01).

**Table 4. E-values for associations between any maternal opioid purchase and adverse neonatal outcomes.**

| Outcome | NICU Admission | LBW | PT | Birth weight z-score | NOWS |
|---|---|---|---|---|---|
| E-Value (95% CI) | 1.34 (1.00-1.74) | 1.51 (1.00-1.86) | 1.53 (1.15-1.86) | 1.34 (1.00–1.50) | 3.63 (2.20-5.68) |

Abbreviations by alphabetical order: CI, confidence interval; LBW, low birth weight; NICU, neonatal intensive care unit; NOWS, neonatal opioid withdrawal syndrome; PT, preterm birth.

For ratio measures, E-values were computed using $E = RR + \sqrt{[RR \times (RR − 1)]}$, where RR is the observed risk or odds ratio. Because all outcomes had a prevalence below 15%, odds ratios were treated as close approximations of risk ratios. For continuous outcomes (birth-weight z-score), standardized mean differences (d) were first converted to risk-ratio equivalents using $RR \approx \exp(0.91 \times |d|)$ prior to applying the E-value formula. When the 95% CI included the null, the lower-bound E-value was set to 1 [36].

**Table 5. Trimester-specific associations between maternal opioid dose (per 100 mg/day MME) and adverse neonatal outcomes.**

| Outcome | Trimester 1 (T1) | Trimester 2 (T2) | Trimester 3 (T3) | T2 vs T1 slope difference (*P*-value) | T3 vs T1 slope difference (*P*-value) |
|---|---|---|---|---|---|
| NICU admission (AOR) | 1.0041 (1.0000-1.0083) | 1.0054 (1.0010-1.0098) | 1.0051 (0.9978-1.0124) | 0.079 | 0.672 |
| LBW (AOR) | 1.0045 (1.0011-1.0080) | 1.0064 (1.0025-1.0103) | 1.0052 (0.9985-1.0120) | 0.106 | 0.764 |
| PT (AOR) | 1.0019 (0.9987-1.0052) | 1.0036 (1.0000-1.0072) | 0.9974 (0.9897-1.0052) | 0.197 | 0.133 |
| Birth-weight z-score (AMD) | −0.0000 (−0.0004, 0.0004) | 0.0000 (−0.0005, 0.0005) | 0.0001 (−0.0006, 0.0008) | 0.548 | 0.476 |
| NOWS (AOR) | 1.0106 (1.0050-1.0162) | 1.0147 (1.0063-1.0231) | 1.0206 (1.0122-1.0291) | 0.198 | 0.0013 |

Abbreviations (alphabetical order): AMD, adjusted mean difference; AOR, adjusted odds ratio; CI, confidence interval; LBW, low birth weight; MME, morphine milligram equivalents; NICU, neonatal intensive care unit; NOWS, neonatal opioid withdrawal syndrome; PT, preterm birth.

**Table 6. Interaction of maternal opioid purchase with benzodiazepine co-Exposure and smoking on adverse neonatal outcomes.**

| Interaction Term | NICU Admission $\chi^2$ (*P*-value) | LBW $\chi^2$ (*P*-value) | PT $\chi^2$ (*P*-value) | Birth-weight z-score $\chi^2$ (*P*-value) | NOWS $\chi^2$ (*P*-value) |
|---|---|---|---|---|---|
| Opioid × Benzodiazepine | 2.01 (0.1560) | 0.46 (0.4960) | 1.52 (0.2180) | 3.78 (0.0520) | 0.02 (0.8840) |
| Opioid × Smoking | 0.60 (0.4390) | 0.07 (0.7850) | 0.56 (0.4550) | 0.89 (0.3450) | 1.79 (0.1810) |

*Abbreviations (alphabetical order):* AMD, adjusted mean difference; Benzo, benzodiazepine; CI, confidence interval; LBW, low birth weight; NICU, neonatal intensive care unit; NOWS, neonatal opioid withdrawal syndrome; PT, preterm birth; SD, standard deviation.

Contrast results (birth-weight z-score model): *Opioid vs No (Benzo = No):* AMD = −0.02 SD (95% CI −0.05 to 0.01; *P* = 0.15); *Opioid vs No (Benzo = Yes):* AMD = −0.17 SD (95% CI −0.32 to −0.02; *P* = 0.02).

No significant opioid × benzodiazepine interactions were detected for other neonatal outcomes. Similarly, no significant interaction was found between opioid exposure and maternal smoking for any of the adverse neonatal outcomes, indicating that the effects of opioid exposure were consistent across smoking status.

## Discussion

This study linked three state data sources (APCD, PDMP, and BCR) and one national data source (SDOH) to characterize opioid purchases, payment methods, and their association with adverse neonatal outcomes among pregnancies in AR. We found that both APCD and PDMP data were essential for comprehensively capturing prescribed opioid purchases, as relying on APCD claims alone would have missed nearly one-fourth of all dispensed opioids. In unadjusted analyses, opioid purchases were associated with higher risks of NICU admission, LBW, PT birth, and lower birth weight z-scores. After adjustment for maternal and neighborhood characteristics, these associations attenuated toward the null, except for a modest but statistically significant association between opioid exposure and PT birth. Opioid purchase remained strongly associated with NOWS even after full adjustment for study covariates. Despite self-paid buyers purchasing substantially higher MME amounts in every trimester compared with insurance-only buyers, no significant differences in neonatal outcomes were observed between the two sub-cohorts after adjusting for study covariates.

Opioids cross the placenta and have been linked to NOWS [38] and, in some studies, to PT birth, LBW, and SGA [3]. Conflicting findings across studies likely reflect residual confounding from maternal socioeconomic and/or clinical factors or confounding by indication, wherein underlying maternal conditions increase both opioid exposure and the risk of poor birth outcomes. A recent systematic review of 80 studies reported associations between in-utero opioid exposure and shorter gestation, LBW, and SGA; however, when lower-quality studies were excluded, these associations weakened, and CI for PT birth and LBW included the null [3]. Similarly, two large population-based studies using approaches to account for both measured and unmeasured confounders found that the apparent adverse effects of opioid exposure were largely attributable to maternal differences [17,39].

Our findings align with these previous findings. After full adjustment, only a modest association with PT birth persisted (AOR 1.14; 95% CI 1.02–1.27), suggesting a small residual effect that may be partially independent of clinical and socio-demographic risk. E-value analysis further supported this interpretation: the E-value for PT birth (1.53; 95% CI 1.15–1.86) indicated modest robustness to unmeasured confounding. In contrast, the larger E-value for NOWS (3.63; 95% CI 2.20–5.68) demonstrated that only a confounder strongly associated with both exposure and outcome could explain away the observed relationship, reinforcing the robustness of the association between prescribed opioid use and NOWS.

With respect to maternal determinants of adverse neonatal outcomes, maternal race identified as non-Hispanic Black was associated with higher odds of PT birth, NICU admission, and lower birth weight, but lower odds of NOWS, compared to non-Hispanic White. These patterns, consistent with previous research [40,41], may reflect underlying disparities in healthcare access and prescribing practices [42].

Evaluation of exposure timing revealed a distinct trimester-specific pattern. Higher opioid intake during the second trimester was associated with higher rates of LBW and NICU admission, whereas third trimester exposure demonstrated the strongest association with NOWS. This pattern aligns with physiological mechanisms as mid-gestation represents a critical period for placental development, during which opioid-mediated alterations in placental perfusion may impair fetal weight gain [43]. By contrast, third-trimester exposure leads to neonatal withdrawal once the placental opioid supply abruptly ceases at birth [44,45]. These findings underscore the importance of simultaneously considering timing and intensity of maternal opioid use when evaluating neonatal risk.

Additional substance exposures may also increase neonatal risk. We observed marginal interaction between concurrent opioid and benzodiazepine exposure in relation to lower birth-weight z-scores, suggesting a possible additive effect on fetal growth restrictions. Similar findings have been reported in prior population-based studies, where prenatal co-exposure to opioids and benzodiazepines was associated with smaller infant size at birth, likely reflecting their combined effects on placental function and fetal metabolism [21,46]. Although this interaction did not extend to other neonatal outcomes in our analysis, it highlights the importance of considering overlapping substance exposures when evaluating perinatal risk among opioid exposed pregnancies.

PDMPs are key tools for monitoring opioid prescribing during pregnancy at a population level, but access is highly restricted due to privacy concerns. Only 16 states, including AR, permit linking PDMP data to patient-level records for research [47]. A Tennessee study linked PDMP to birth records but emphasized the need for additional administrative data to better capture maternal diagnoses and out-of-state prescriptions [48]. Our study was novel in linking APCD, PDMP, BCR, and SDOH data sources. This comprehensive data integration enhanced the accuracy of opioid exposure assessment and enabled a more robust analysis of adverse neonatal outcomes.

We acknowledge that randomized controlled trials are the gold standard for causal inference; however, assigning pregnant women to opioid use versus non-use would be unethical. As a result, our understanding of patterns and effects of opioid exposure during pregnancy must rely on observational data like ours, which has limitations. First is the inability to confirm actual opioid consumption; while this study documents purchase patterns; it cannot verify whether pregnant women ingested the medications. Though, medication purchases are accepted as a valid adherence measure by the Centers for Medicare and Medicaid Services, particularly when using large administrative databases [49,50]. Second is

the potential exposure misclassification, as illicit opioid purchases cannot be captured. This misclassification would likely bias associations toward the null, meaning the true impact of total opioid exposure on neonatal outcomes may be underestimated when focusing on prescribed opioids. Accordingly, the small number of NOWS cases in the unexposed group in our study may reflect undetected illicit use or alternatively withdrawal from non-opioid drugs. Despite these limitations, our study has key strengths: we linked multiple data sources for comprehensive exposure assessment of prescribed opioids, adjusted for a wide range of covariates, and found that after adjusting for key characteristics, the results were largely unchanged using alternative model specifications. Additionally, our use of MME improved the precision of exposure to prescribed opioids.

## Conclusion

Our study demonstrated a robust association between prescribed opioid purchase and NOWS, a modest but significant association with PT birth, but no independent associations with NICU admission, LBW, or fetal growth after multivariable adjustment. These findings suggest that much of the observed relationship between prenatal opioid exposure and adverse neonatal outcomes is driven by underlying maternal health and social characteristics rather than opioid use alone. The timing and intensity of opioid exposure also appear to be important, with higher third-trimester doses showing the strongest association with NOWS. Exploratory analyses further emphasized the relevance of co-exposures to psychoactive medications, particularly benzodiazepines, which demonstrated marginal interaction with opioids and may contribute to fetal growth restriction. Notably, self-paid opioid purchases did not confer additional risk of adverse neonatal outcomes after adjusting for study covariates. Lastly, our findings underscore the importance of capturing opioid exposure from multiple data sources, as studies relying on administrative claims exclusively may appreciably understate prescribed opioid purchases among pregnant women.

## Supporting information

**S1 Appendix. Data sources and definitions of exposure, outcome, and covariate measures.**
(DOCX)

**S1 Table. Comparison of maternal neighborhood characteristics between the study cohorts and sub-cohorts.**
(DOCX)

**S2 Table. Summary of results of models evaluating the association between study covariates and neonatal outcomes.**
(DOCX)

**S3 Table. Summary of results of fully adjusted models evaluating association between study covariates and neonatal outcomes.**
(DOCX)

## Author contributions

**Conceptualization:** Nahed O. ElHassan, Bradley C. Martin.

**Data curation:** Corey J. Hayes.

**Formal analysis:** Nahed O. ElHassan, Xiaotong Han, Chary Akmyradov.

**Funding acquisition:** Corey J. Hayes.

**Investigation:** Ruchira V. Mahashabde, Chenghui Li, Teresa Hudson, Robert Mcgehee Jr, Peter M. Mourani, Bradley C. Martin.

**Methodology:** Nahed O. ElHassan, Bradley C. Martin.

**Supervision:** Bradley C. Martin.

**Writing – original draft:** Nahed O. ElHassan.

**Writing – review & editing:** Nahed O. ElHassan, Corey J. Hayes, Ruchira V. Mahashabde, Xiaotong Han, Chary Akmyradov, Chenghui Li, Teresa Hudson, Robert Mcgehee Jr, Peter M. Mourani, Bradley C. Martin.

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
