## [Decision Letter · Decision Letter 0]

3 Oct 2025

Dear Dr. Martin,

Thank you for submitting your manuscript to PLOS ONE. After careful consideration, we feel that it has merit but does not fully meet PLOS ONE’s publication criteria as it currently stands. Therefore, we invite you to submit a revised version of the manuscript that addresses the points raised during the review process.

We look forward to receiving your revised manuscript.

Kind regards,

Keith Anthony Dookeran, MD PhD

Academic Editor

PLOS ONE

Journal Requirements:

[Research reported in this publication was supported by the National Center for Advancing Translational Sciences of the National Institutes of Health under award number UL1 TR003107. The content is solely the responsibility of the authors and does not necessarily represent the official views of the National Institutes of Health.].

3. Thank you for stating the following in your manuscript:

[Research reported in this publication was supported by the National Center for Advancing Translational Sciences of the National Institutes of Health under award number UL1 TR003107. The content is solely the responsibility of the authors and does not necessarily represent the official views of the National Institutes of Health.]

[Research reported in this publication was supported by the National Center for Advancing Translational Sciences of the National Institutes of Health under award number UL1 TR003107. The content is solely the responsibility of the authors and does not necessarily represent the official views of the National Institutes of Health.]

[Dr. Martin receives royalties from TrestleTree LLC for the commercialization of an opioid risk prediction tool which is unrelated to the current investigation. The rest of the co-authors have no financial relationship relevant to this presentation to disclose. All other authors have no conflict of interest to disclose.].

We note that you received funding from a commercial source: TrestleTree LLC

5. In the online submission form, you indicated that [This project is an analysis of existing health records. Access to these databases was obtained through a data use agreement (DUA) from the Arkansas Insurance Department and Arkansas Center for Health Improvement (ACHI). The DUA does not permit sharing of data and individuals interested in acquiring the data can make a request to obtain the data from ACHI.].

Reviewers' comments:

Reviewer's Responses to Questions

**Comments to the Author**

1. Is the manuscript technically sound, and do the data support the conclusions?

Reviewer #1: Yes

Reviewer #2: Yes

2. Has the statistical analysis been performed appropriately and rigorously?

Reviewer #1: I Don't Know

Reviewer #2: No

3. Have the authors made all data underlying the findings in their manuscript fully available?

Reviewer #1: Yes

Reviewer #2: Yes

4. Is the manuscript presented in an intelligible fashion and written in standard English?

Reviewer #1: Yes

Reviewer #2: Yes

Reviewer #1: Dear Editors, Dear Authors,

Thank you for the opportunity to read and review this manuscript.

This scientific paper presents the results of a longitudinal retrospective cohort analysis aimed at evaluating patterns of opioid purchases by payment source among pregnant women in Arkansas and examining their associations with adverse neonatal outcomes. Multiple data sources were utilized and appropriately linked to address clearly defined hypotheses. The databases and linkage methods are well described and transparent.

The manuscript is well written, and the discussion provides valuable insights in the context of the existing literature.

While I do not consider myself qualified to evaluate the statistical robustness of the analyses, from a clinical and scientific standpoint, I have no modifications to suggest.

Reviewer #2: The premise for the current study is that in the context of assessing neonatal outcomes following maternal opioid use, prior studies likely underestimate actual opioid exposure and fail to evaluate differences in maternal characteristics between opioid users and non-users.

Hence, the study objective is to evaluate patterns of opioid purchases among pregnant women in Arkansas and examine their associations with adverse neonatal outcomes.

This is regarded as a worthy topic for research.

I recommend revise and re-submit.

The study leverages a linkage between a statewide claims database [the Arkansas All-Payer Claims Database (AR-APCD), and the Arkansas Prescription Drug Monitoring Programs (AR-PDMP)] from 2014-2016 to characterize opioid prescription purchases among pregnant women and evaluate their impact on adverse neonatal outcomes.

The authors hypothesize that pregnant women who self-paid for opioid analgesics, potentially to bypass payer restrictions or detection, represent a higher-risk group due to higher opioid purchase, and that both opioid purchase and opioid amount are associated with increased risk of adverse neonatal outcomes (i.e., preterm birth, lower birth weight, NICU admission, and NOWS).

The study design is a longitudinal retrospective cohort analysis of singleton pregnancies and in general is well conducted but there are some issues that need attention.

Comments:

1. The authors should clearly state the inclusion and exclusion criteria in the methods section with any necessary justification. Regarding Figure 1 and sample exclusion criteria, the authors do not explain why they chose to exclude 40,340 pregnancies where mothers ‘did not have continuous medical and pharmacy benefits.’ This is not explained in the methods and is considered a deficiency. This also seems curious as major exposures are simply classified as opioid self-paid buyers, opioid insurance-only buyers, and non-buyers, and presumably, mothers did not need to have continuous medical and pharmacy benefits to be categorized as buyers. Further, it would be interesting to know whether results of the study would be different had these pregnancies not been excluded, so justification to exclude is required or alternatively, the authors could do a sensitivity analysis including these mothers.

2. There is no mention of exploration for model interaction/effect modification in the methods section, and the authors need to comment on how this was approached and whether explored.

3. Lines 259-260 seems incorrect as it states, ‘Table 3 presents the AORs from sequential regression models comparing opioid buyers to non-buyers, insurance-only buyers to non-buyers, and self-paid buyers to insurance-only buyers’ but Table 3 does not show results from models comparing ‘any opioid buyers to non-buyers.’ Regarding the choice of groups for comparison using regression models (specifically Table 3), it would seem that it would be of interest to show ‘any opioid buyer vs. non-buyers.’ Hence, I suggest that the authors revise Table 3 to only show rows for Model 1 and 8 (i.e., the crude and fully adjusted models; the other models can go into the supplement, and also interesting to note that estimates for models 5 through 8 are fairly similar, and models 7 and 8 are nearly identical) and add the comparison of ‘any opioid buyer vs. non-buyers.’

4. Another issue with Table 3 is the strange estimates for the comparison of opioid self-paid vs. insurance-only buyers. As the estimates for opioid self-paid vs. the common referent appears to be consistently larger than for insurance-only buyers, one would expect to see coefficients larger than the null for the comparison of opioid self-paid vs. insurance-only buyers, but regardless of model or outcome, these estimates appear inverted which is non-intuitive and likely due to issues with multicollinearity or perhaps non-collapsibility as estimates are ORs. Although these estimates are equivocal, the authors should explain and, also suggest that these could be moved to supplement as largely non-informative.

5. Given that models 5 through 8 show fairly similar results, suggest that the authors undertake a sensitivity analysis to address unmeasured confounding through the “E-value” method, described in VanderWeele TJ, Ding P. Sensitivity analysis in observational research: Introducing the “E-Value.” Annals of Internal Medicine 2017;167(4):268-274.

6. Regarding the mediation analyses, the authors do not provide adequate methodological details. For example, what modeling approach is being used, what assumptions are applied, and could there be any exposure mediator-interaction? [For further information please see: VanderWeele TJ. Mediation Analysis: A Practitioner's Guide. Annu Rev Public Health. 2016;37:17-32. doi: 10.1146/annurev-publhealth-032315-021402. Epub. 2015 Nov 30. PMID: 26653405]. In addition, the results shown in Table 4 demonstrate that estimates overall (i.e., the total effects) are largely null. Hence, the authors’ discussion on mediation indirect effects are more consistent with a theoretical statistical approach, rather than an epidemiological approach, as with the latter, there would be no point in disaggregating a null effect. Hence, suggest that the authors omit the mediation analysis in any revision.

**Do you want your identity to be public for this peer review?** For information about this choice, including consent withdrawal, please see our Privacy Policy

Reviewer #1: **Yes:** Dr Anastasia DEMINA

Reviewer #2: No

---

## [Author Response · Author response to Decision Letter 1]

25 Nov 2025

Keith Anthony Dookeran, MD, PhD

Manuscript Title:

“Novel approaches in linkage of data sources to explore the associations between purchase of opioid prescriptions during pregnancy and adverse neonatal outcomes”

Manuscript ID:

PONE-D-25-27057

Dear Dr. Dookeran,

We thank you and the Reviewers for the feedback on our manuscript. We have carefully revised the manuscript and associated documents in accordance with all recommendations. Below, we provide a detailed, point-by-point response to each comment, including updated statements where applicable. A tracked and untracked version of the revised manuscript has been submitted.

1. PLOS ONE style requirements and file naming

Editor’s Request: Ensure the manuscript meets PLOS ONE style and file-naming requirements.

Response:

We have reviewed the PLOS ONE style guidelines and updated the manuscript and all associated files accordingly. File names, figure labels, and formatting now conform fully to PLOS ONE requirements.

2. Revised Funding Statement

Editor’s Request: Provide a Funding Statement that declares all sources of support and includes the required sentence, “There was no additional external funding received for this study.”

Response:

We have updated the Funding Statement to explicitly declare all sources of support. The only support for this study was the NCATS/NIH award UL1 TR003107. The required sentence has been added. The revised Funding Statement is:

Revised Funding Statement:

All funding for this study was provided by the National Center for Advancing Translational Sciences of the National Institutes of Health under Award Number UL1 TR003107. The content is solely the responsibility of the authors and does not necessarily represent the official views of the National Institutes of Health. There was no additional external funding received for this study.

3. Removal of funding text from manuscript

Editor’s Request: Remove funding details from the Acknowledgments or elsewhere; maintain funding statements only in the Funding Statement section.

Response:

We have removed all funding-related text from the manuscript, including from the Acknowledgments section. The Funding Statement above has been updated and included as above in the cover letter.

4. Revised Competing Interests Statement

Editor’s Request: Explicitly state the commercial entity (TrestleTree LLC) related to Dr. Martin and include any restrictions on data sharing, along with the statement required by PLOS ONE.

Response:

We have updated the Competing Interests Statement to clearly identify TrestleTree LLC and to outline data-sharing restrictions required by state law and our DUA. The revised statement is provided below.

Revised Competing Interests Statement:

Dr. Martin receives royalties from TrestleTree LLC for the commercialization of an opioid risk prediction tool, which is unrelated to the current investigation and hence TrestleTree played no role in this manuscript. The remaining authors declare no financial relationships, employment, consultancy, patents, products in development, or marketed products relevant to this work .

There are restrictions on data and material sharing due to state and federal privacy protections governing the Arkansas All-Payer Claims Database (APCD). As stipulated by the data use agreement with the Arkansas Center for Health Improvement (ACHI), the authors are not permitted to share raw or de-identified APCD data. All publications and presentations using APCD data must undergo review by the Arkansas Insurance Department and the Healthcare Transparency Initiative Board prior to submission. Interested researchers may request access to APCD data for their own approved projects by following the formal request procedures outlined at www.arkansasapcd.net.

This does not alter our adherence to PLOS ONE policies on sharing data and materials.

5. Data Availability and request for exemption

Editor’s Request: Address PLOS ONE’s data-sharing policy and provide justification if data cannot be publicly shared.

Response:

The APCD is governed by state and federal privacy protections under the Arkansas Healthcare Transparency Initiative. Access is strictly regulated via a Data Use Agreement with ACHI and the Arkansas Insurance Department. Sharing raw or de-identified APCD data publicly would violate these legal requirements and the DUA.

Although the dataset cannot be deposited in a public repository, APCD data remain accessible to qualified researchers through ACHI. Requests can be made via www.arkansasapcd.net and are subject to regulatory and privacy review.

We respectfully request an exemption from public data deposition for legal and ethical reasons. All analytic procedures, code descriptions, and variable definitions are fully described in the manuscript to support reproducibility for researchers who obtain their own approved access to APCD data.

Revised Data Sharing Statement:

Data used in this study were obtained from the Arkansas All-Payer Claims Database (APCD), which is governed by state and federal privacy protections and cannot be shared publicly under the terms of the Data Use Agreement with the Arkansas Center for Health Improvement (ACHI) and the Arkansas Insurance Department. Interested researchers may request access to APCD data for approved research projects through the formal request process at www.arkansasapcd.net. All analytic procedures, variable definitions, and modeling specifications are provided in the manuscript to support reproducibility.

Responses to Reviewers

Reviewer 1

Comment 1:

Dear Editors, Dear Authors,

Thank you for the opportunity to read and review this manuscript.

This scientific paper presents the results of a longitudinal retrospective cohort analysis aimed at evaluating patterns of opioid purchases by payment source among pregnant women in Arkansas and examining their associations with adverse neonatal outcomes. Multiple data sources were utilized and appropriately linked to address clearly defined hypotheses. The databases and linkage methods are well described and transparent. The manuscript is well written, and the discussion provides valuable insights in the context of the existing literature.

While I do not consider myself qualified to evaluate the statistical robustness of the analyses, from a clinical and scientific standpoint, I have no modifications to suggest.

Response:

We appreciate the positive feedback regarding the clarity of the research questions, the transparency of the data linkage methods, and the clinical relevance of the findings.

Reviewer 2

Comment 1:

The authors should clearly state the inclusion and exclusion criteria in the methods section with any necessary justification. Regarding Figure 1 and sample exclusion criteria, the authors do not explain why they chose to exclude 40,340 pregnancies where mothers ‘did not have continuous medical and pharmacy benefits.’ This is not explained in the methods and is considered a deficiency. This also seems curious as major exposures are simply classified as opioid self-paid buyers, opioid insurance-only buyers, and non-buyers, and presumably, mothers did not need to have continuous medical and pharmacy benefits to be categorized as buyers. Further, it would be interesting to know whether results of the study would be different had these pregnancies not been excluded, so justification to exclude is required or alternatively, the authors could do a sensitivity analysis including these mothers.

Response:

We have revised the Methods section and the footnote of Fig 1 to clearly describe all inclusion and exclusion criteria and to justify the requirement for continuous medical and pharmacy coverage.

Continuous enrollment from six months before conception through delivery was required to ensure complete and accurate capture of maternal opioid purchases, other prescription medications, and clinical comorbidities recorded in claims. Intermittent or partial coverage would lead to incomplete claims capture and a substantial risk of misclassification of both exposures and covariates. This approach aligns with established pharmacoepidemiologic guidance recommending continuous enrollment to minimize exposure misclassification and confounding when using administrative claims data (Reference: Schneeweiss S, Avorn J. A review of uses of health care utilization databases for epidemiologic research. J Clin Epidemiol. 2005;58(4):323-37. doi: 10.1016/j.jclinepi.2004.10.012.)

Because pregnancies without continuous coverage would have incomplete ascertainment of opioid purchases and key covariates, including these individuals would potentially introduce differential misclassification and bias. Consistent with prior claims-based opioid research, we retained the continuous enrollment requirement for the primary analysis and did not perform a sensitivity analysis including these pregnancies.

Comment 2:

There is no mention of exploration for model interaction/effect modification in the methods section, and the authors need to comment on how this was approached and whether explored.

Response:

We appreciate this suggestion as it prompted us to explore potential effect modification and have now revised the Methods section to explicitly describe how interaction and effect modification were evaluated.

Specifically, we assessed whether the association between opioid exposure and neonatal outcomes varied by dose or timing by incorporating an exposure × morphine milligram equivalent (MME) × trimester interaction term, which enabled evaluation of trimester-specific effects per 100 mg/day MME. In addition, as a post hoc exploratory analysis, we examined potential effect modification by maternal smoking and concurrent benzodiazepine exposure using product (interaction) terms (e.g., exposure × smoking; exposure × benzodiazepine).

These details have been added to the Methods section for clarity, and corresponding results and discussion of interaction findings have been incorporated into the Results and Discussion sections.

Comment 3:

Lines 259-260 seem incorrect as it states, ‘Table 3 presents the AORs from sequential regression models comparing opioid buyers to non-buyers, insurance-only buyers to non-buyers, and self-paid buyers to insurance-only buyers’ but Table 3 does not show results from models comparing ‘any opioid buyers to non-buyers.’ Regarding the choice of groups for comparison using regression models (specifically Table 3), it would seem that it would be of interest to show ‘any opioid buyer vs. non-buyers.’ Hence, I suggest that the authors revise Table 3 to only show rows for Model 1 and 8 (i.e., the crude and fully adjusted models; the other models can go into the supplement, and also interesting to note that estimates for models 5 through 8 are fairly similar, and models 7 and 8 are nearly identical) and add the comparison of ‘any opioid buyer vs. non-buyers.’

Response:

We have revised the Statistical Analysis section and Table 3 to now include a comparison of any opioid buyer versus non-buyer, consistent with the study’s primary exposure contrast.

To enhance clarity and streamline the presentation, we have followed the reviewer’s suggestion to present only Model 1 (unadjusted) and Model 8 (fully adjusted) in the main manuscript. The full sequence of models (Models 1–8) is now provided in the Supplemental Materials for transparency. Table 3 has been updated accordingly, and the text in the Results section has been corrected to align with these revisions.

Comment 4:

Another issue with Table 3 is the strange estimates for the comparison of opioid self-paid vs. insurance-only buyers. As the estimates for opioid self-paid vs. the common referent appears to be consistently larger than for insurance-only buyers, one would expect to see coefficients larger than the null for the comparison of opioid self-paid vs. insurance-only buyers, but regardless of model or outcome, these estimates appear inverted which is non-intuitive and likely due to issues with multicollinearity or perhaps non-collapsibility as estimates are ORs. Although these estimates are equivocal, the authors should explain and, also suggest that these could be moved to supplement as largely non-informative.

Response:

We appreciate the reviewer’s comments regarding the non-intuitive estimates for the comparison of opioid self-paid versus insurance-only buyers. As suggested, we examined potential methodological explanations for this pattern.

First, we formally evaluated multicollinearity using variance inflation factors (VIFs) and tolerance statistics. All retained covariates had VIF values <5, consistent with accepted thresholds indicating acceptable collinearity. The Statistical Analysis and Results sections have been updated to reflect completion of the multicollinearity assessment and the revised regression models after removal of any variables identified as highly collinear.

We agree that the observed pattern is more consistent with non-collapsibility of the odds ratio, whereby adjusted odds ratios can diverge from unadjusted estimates even in the absence of confounding, particularly when baseline risks vary across exposure groups.

Given that the self-paid versus insurance-only comparison yielded largely non-informative and inconsistent patterns across sequential models, and that Models 2 through 7 added limited interpretive value, we have moved these intermediate models to Supplemental Materials. The main manuscript now presents only the unadjusted (Model 1) and fully adjusted (Model 8) results to improve clarity and emphasize the most clinically meaningful contrasts.

Comment 5:

Given that models 5 through 8 show fairly similar results, suggest that the authors undertake a sensitivity analysis to address unmeasured confounding through the “E-value” method, described in VanderWeele TJ, Ding P. Sensitivity analysis in observational research: Introducing the “E-Value.” Annals of Internal Medicine 2017;167(4):268-274.

Response:

We thank the reviewer for this excellent suggestion. We have now incorporated an E-value sensitivity analysis to assess the robustness of our findings to potential unmeasured confounding. The Statistical Analysis section has been updated to indicate that E-values were calculated for the primary exposure–outcome comparisons (any opioid purchaser vs. non-purchaser). Corresponding E-value results have been added to the Results section, and a brief interpretation has been incorporated into the Discussion to contextualize the findings. These revisions clarify the extent to which unmeasured confounding would be required to fully explain the observed associations.

Comment 6:

Regarding the mediation analyses, the authors do not provide adequate methodological details. For example, what modeling approach is being used, what assumptions are applied, and could there be any exposure mediator-interaction? [For further information please see: VanderWeele TJ. Mediation Analysis: A Practitioner's Guide. Annu Rev Public Health. 2016;37:17-32. doi: 10.1146/annurev-publhealth-032315-021402. Epub. 2015 Nov 30. PMID: 26653405]. In addition, the results shown in Table 4 demonstrate that estimates overall (i.e., the total effects) are largely null. Hence, the authors’ discussion on mediation indirect effects are more consistent with a theoretical statistical approach, rather than an epidemiological approach, as with the latter, there would be no point in disaggregating a null effect. Hence, suggest that the authors omit the mediation analysis in any revision.

Response:

We agree that the mediation analyses, as originally presented, were largely theoretical to better understand the relationships between the source of payment, opioid dose and our study outcomes and agree that disaggregating indirect effects in the absence of meaningful total effects i

---

## [Decision Letter · Decision Letter 1]

29 Dec 2025

Novel approaches in linkage of data sources to explore the associations between purchase of opioid prescriptions during pregnancy and adverse neonatal outcomes

PONE-D-25-27057R1

Dear Dr. Martin,

We’re pleased to inform you that your manuscript has been judged scientifically suitable for publication and will be formally accepted for publication once it meets all outstanding technical requirements.

Kind regards,

Keith Anthony Dookeran, MD PhD

Academic Editor

PLOS One

Additional Editor Comments (optional):

Reviewers' comments:

Reviewer's Responses to Questions

**Comments to the Author**

Reviewer #2: All comments have been addressed

2. Is the manuscript technically sound, and do the data support the conclusions?

Reviewer #2: Yes

3. Has the statistical analysis been performed appropriately and rigorously?

Reviewer #2: Yes

4. Have the authors made all data underlying the findings in their manuscript fully available?

Reviewer #2: No

5. Is the manuscript presented in an intelligible fashion and written in standard English?

Reviewer #2: Yes

Reviewer #2: (No Response)

**Do you want your identity to be public for this peer review?** For information about this choice, including consent withdrawal, please see our Privacy Policy

Reviewer #2: No

---

## [Editor Report · Acceptance letter]

PONE-D-25-27057R1

PLOS One

Dear Dr. Martin,

I'm pleased to inform you that your manuscript has been deemed suitable for publication in PLOS One. Congratulations! Your manuscript is now being handed over to our production team.

Kind regards,

on behalf of

Dr. Keith Anthony Dookeran

Academic Editor

PLOS One